# Biomimetic 3D Models for Investigating the Role of Monocytes and Macrophages in Atherosclerosis

**DOI:** 10.3390/bioengineering7030113

**Published:** 2020-09-16

**Authors:** Anna Garcia-Sabaté, Walaa Kamal E. Mohamed, Jiranuwat Sapudom, Aseel Alatoom, Layla Al Safadi, Jeremy C. M. Teo

**Affiliations:** 1Laboratory for Immuno Bioengineering Research and Applications, Division of Engineering, New York University Abu Dhabi, 129188 Abu Dhabi, UAE; anna.sabate@nyu.edu (A.G.-S.); wm1081@nyu.edu (W.K.E.M.); jiranuwat.sapudom@nyu.edu (J.S.); aseel.alatoom@nyu.edu (A.A.); las892@nyu.edu (L.A.S.); 2Departament de Genètica i Microbiologia, Facultat de Biociències, Universitat Autònoma de Bellaterra, 08193 Barcelona, Spain; 3Department of Mechanical and Biomedical Engineering, Tandon School of Engineering, New York University, New York, NY 11201, USA

**Keywords:** atherosclerosis, monocyte, macrophage, disease model, collagen, 3D cell culture, immunomechanobiology

## Abstract

Atherosclerosis, the inflammation of artery walls due to the accumulation of lipids, is the most common underlying cause for cardiovascular diseases. Monocytes and macrophages are major cells that contribute to the initiation and progression of atherosclerotic plaques. During this process, an accumulation of LDL-laden macrophages (foam cells) and an alteration in the extracellular matrix (ECM) organization leads to a local vessel stiffening. Current in vitro models are carried out onto two-dimensional tissue culture plastic and cannot replicate the relevant microenvironments. To bridge the gap between in vitro and in vivo conditions, we utilized three-dimensional (3D) collagen matrices that allowed us to mimic the ECM stiffening during atherosclerosis by increasing collagen density. First, human monocytic THP-1 cells were embedded into 3D collagen matrices reconstituted at low and high density. Cells were subsequently differentiated into uncommitted macrophages (M0) and further activated into pro- (M1) and anti-inflammatory (M2) phenotypes. In order to mimic atherosclerotic conditions, cells were cultured in the presence of oxidized LDL (oxLDL) and analyzed in terms of oxLDL uptake capability and relevant receptors along with their cytokine secretomes. Although oxLDL uptake and larger lipid size could be observed in macrophages in a matrix dependent manner, monocytes showed higher numbers of oxLDL uptake cells. By analyzing major oxLDL uptake receptors, both monocytes and macrophages expressed lectin-like oxidized low-density lipoprotein receptor-1 (LOX1), while enhanced expression of scavenger receptor CD36 could be observed only in M2. Notably, by analyzing the secretome of macrophages exposed to oxLDL, we demonstrated that the cells could, in fact, secrete adipokines and growth factors in distinct patterns. Besides, oxLDL appeared to up-regulate MHCII expression in all cells, while an up-regulation of CD68, a pan-macrophage marker, was found only in monocytes, suggesting a possible differentiation of monocytes into a pro-inflammatory macrophage. Overall, our work demonstrated that collagen density in the plaque could be one of the major factors driving atherosclerotic progression via modulation of monocyte and macrophages behaviors.

## 1. Introduction

Atherosclerosis is the dominant underlying causation of coronary heart disease and cerebrovascular disease [1] and statistically a major cause of morbidity and mortality worldwide [2,3]. The systematic formation of detrimental atherosclerotic plaques, via monocyte to macrophage differentiation to foam cell formation, which eventually narrows and occludes arteries, is well-reviewed elsewhere [4,5]. In short, monocytes are recruited to lesion sites, where low-density lipoproteins (LDL) and apolipoprotein B-containing lipoproteins have accumulated and activated the endothelium. Monocytes then infiltrate the lesion and differentiate into macrophages, which will uptake lipoproteins, ultimately resulting in foam cells that comprise the central core of atheromas that will progressively occlude the vessel. Atherosclerosis is considered a non-resolving inflammatory condition [2,6], akin to chronic wounds that will not resolve towards healing. On account of this close association with wound healing, and evidence that atherosclerotic plaques are vastly populated by pan-macrophage marker CD68+ cells, macrophages are thought to play other lead roles beyond foam cell formation that progresses plaque buildup [7,8].

In atherosclerosis, macrophages are implicated in progression or regression of plaques, as well as in plaque stabilization or its rupture [9,10]. Pro-inflammatory macrophages, or M1 macrophages, are responsible for progression, leading towards instability and finally resulting in rupture [11]. Conventionally, at the complete opposite end of the macrophage spectrum, sits the anti-inflammatory or M2 macrophages [12], which mediate a more favorable outcome of atherosclerosis [13]. It is now accepted that macrophage phenotypes are not dichotomous but lie along this M1-M2 spectrum, and depending on the atherosclerotic cellular niche, macrophages can dynamically traverse this M1-M2 continuum [12,14], affecting the outcome of this disease. While significant advancements have been made in terms of disease management, research must now shift towards understanding the mechanisms of atherosclerosis regression and the repair of atherosclerotic lesions [15]. It is, therefore, imperative to understand macrophage biology in the interest of this new pivot against atherosclerosis.

Animal models have provided abundant information for atherosclerosis research, however, to understand atherosclerosis healing mechanistically, relevant in vitro models are a must. Models of atherosclerosis have progressed from simple 2D culture to 3D multi-cellular cultures [16,17], the latter a better representation considering the dimensionality of the disease and anatomical features involved. To mimic in vivo-like conditions, in vitro models have encapsulated the relevant cells of interest within naturally-derived and polymer-based hydrogels.

Several 3D models of atherosclerosis have been published [18,19,20,21]. While they mostly tackled the interaction between different cells in atherosclerotic plaques at different stages of plaque progression, they overlooked some factors which we aim to address. The 3D extracellular matrix (ECM) used for scaffolding the models was not assessed in terms of its biomechanical influence on monocytes and macrophages in regard to atherosclerosis’ cellular niche. This is especially important since collagen comprises up to 60% of the plaque protein [22], and the amount of collagen in the plaque dictates its mechanical stability [23], where a deficiency of collagen may lead to plaque rupture and an excess results in narrowing of the artery. Moreover, aged or obese individuals are known to exhibit notably stiffened and thickened arteries [24]. Also, our previous work has specifically shown that macrophage functionality and immune phenotype is regulated by physical parameters of the surrounding ECM [25]. While the majority of biological readouts from 2D and 3D in vitro atherosclerosis models have focused on LDL uptake by the immune cells and secretion of specific pro-inflammatory cytokines, there is a lack of comprehensive secretome studies. Owing to the complexity of the biochemical milieu in atherosclerotic microenvironments, we want to examine adipokines and growth factors, two groups of cytokines that are currently vague, but crucial to understanding the mechanisms of regression and repair of atherosclerotic plaques. Furthermore, most 3D models commonly employ monocytes or resident macrophages (M0) for LDL exposure to simulate atherosclerosis. However, there are pathological situations, for example, dyslipidemia, diabetes, hypertension, obesity, and smoking, whereby transendothelial permeability to LDLs increases [26,27] and resident macrophages of varying phenotypes within the intima could uptake LDLs, thus also modulate atherosclerotic plaque [14].

In this study, we employ a system of collagen-based hydrogels with different matrix densities that recapitulate the porosity of early (low tissue density) and advanced (high tissue density) atherosclerotic tissue in vivo [28,29]. Within these fibrillar collagen hydrogels, a commonly used and genetically uniform human monocytic cell line, THP-1, is embedded, differentiated into various macrophage phenotypes, and later exposed to oxidized LDLs (oxLDL). Here, we aim to elucidate the effects of the predominant ECM, collagen, on these immune cells by isolating them from other factors such as different cell types or proteins. The assessment was performed to evaluate oxLDL uptake, scavenger receptors and immune phenotype gene expressions, and their secretions of adipokine and growth factors.

## 2. Materials and Methods

### 2.1. Cell Culture

Human monocytic cell line THP-1 were maintained in RPMI-1640 (Gibco, Thermo Fisher Scientific, Inc., Waltham, MA, USA) with 10% fetal bovine serum (FBS), 1% HEPES, 1% sodium pyruvate, 0.01% beta-mercaptoethanol, and 1% penicillin/streptomycin (all from Invitrogen, Thermo Fisher Scientific, Inc.) at 37 °C, 95% humidity and 5% CO_2_ (standard cell culture conditions).

### 2.2. Embedding and Differentiating THP-1 towards Macrophages in 3D Collagen Matrices

3D collagen matrices at concentrations of 1 and 3 mg/mL were prepared as previously published [25,30]. In short, a collagen solution was prepared by mixing of type I rat tail collagen (Advanced BioMatrix, Inc., Carlsbad, CA, USA) with 250 mM of phosphate buffer and 0.1% acetic acid (both from Sigma-Aldrich, Inc., St. Louis, MO, USA). Afterward, 1 × 10^5^ THP-1 cells were suspended in the prepared collagen solution and then transferred onto glutaraldehyde-coated coverslips. Cell-free collagen matrices were analyzed prior to use for cell culture regarding their topological and mechanical properties, using an image-based analysis toolbox [31] and non-destructive rheometer [25,32], respectively.

Macrophage differentiation was performed using an established protocol [25]. In brief, THP-1 cells were differentiated into uncommitted macrophages (M0) by culturing in RPMI 1640 media without FBS supplemented with 300 nM phorbol 12-myristate 13-acetate (PMA; Sigma-Aldrich, Inc.) for 6 h. To achieve pro-inflammatory (M1) and anti-inflammatory (M2) macrophages, PMA containing media was removed, and M0 were rested in RPMI-1640 cell culture media without FBS for 24 h. Afterward, cells were cultured in activating media for 48 h. For M1 activation, RPMI-1640 cell culture media without FBS was supplemented with 10 pg/mL lipopolysaccharide (LPS, Sigma-Aldrich, Inc.) and 20 ng/mL interferon-gamma IFN-γ (Biolegend, Inc., San Diego, CA, USA), while 20 ng/mL interleukin 4 (IL-4, Biolegend, Inc.) and 20 ng/mL interleukin 13 (IL-13, Biolegend, Inc.) was used for M2 activation.

### 2.3. Topological and Mechanical Characterization of 3D Collagen Matrices

Cell-free 3D collagen matrices were analyzed regarding their topological and mechanical properties based on previously published protocols [32]. In brief, 5- (and-6)-carboxytetramethylrhodaminesuccinimidylester (5(6)-TAMRA-SE) (Sigma-Aldrich, Inc.) stained 3D collagen matrices were imaged using confocal microscopy (SP8; Leica Microsystems GmbH, Wetzlar, Germany) at 63X. *Z*-stacked (50 µm depth, at 5 µm intervals) image datasets (of pixel size 0.13 × 0.13 µm) were analyzed using a custom-built image analysis toolbox [31] to obtain mean pore size. For each type of 3D collagen matrix, analysis was performed in triplicate with three positions per sample. Measurements of mechanical properties was performed on cell-free label-free 3D collagen matrices non-destructively (ElastoSense™ Bio; Rheolution Inc., Montreal, QC, Canada). Briefly, fibrillation of 3 mL of collagen solution was initiated at 37 °C and the elastic modulus of collagen was measured during polymerization. Three samples per matrix type were analyzed.

### 2.4. Treatment with oxLDL

Monocyte (THP-1), plus uncommitted and activated macrophages were cultured in the presence of 5 µg/mL oxidized low-density lipoproteins from human plasma (oxLDL; Molecular Probes™, Thermo Fisher Scientific, Inc.) for 5 days at standard cell culture condition. oxLDL conjugated 1,1′-dioctadecyl-3,3,3′,3′-tetramethylindocarbocyanine perchlorate (DiI) was used for quantification and visualization of LDL uptake. As a control, cells were cultured at similar conditions without oxLDL.

### 2.5. Quantitative Analysis of oxLDL Uptake

After 5 days of LDL treatment, collagen matrices were digested with 6 mg/mL of collagenase (Worthington Biochemical Corp., Lakewood, NJ, USA) prepared in cell culture media for 15 min at standard cell culture conditions to digest the collagen. Afterward, cells were analyzed using Attune NxT Flow Cytometer equipped with autosampler (Thermo Fisher Scientific, Inc.). Percentage of oxLDL uptake cells and mean fluorescent intensity were quantified using FlowJow Software (Build 10.5.3; BD Life Sciences). Experiments were performed in 5 replicates.

### 2.6. Quantitative Analysis of Lipid Droplet Size

For analysis of lipid droplet size, cells were first fixed with 4% paraformaldehyde and stained with 4′,6-diamidino-2-phenylindole (DAPI; 1:10,000 dilution in PBS) for 2 h at room temperature. Afterward, cells were imaged using a confocal laser scanning microscope (cLSM) (SP8; Leica Microsystems GmbH). The cLSM stacked images were gathered and analyzed regarding the lipid size using DiI-oxLDL fluorescence signal by a home-built MATLAB script (MATLAB 2019a; The MathWorks, Inc., Natick, MA, USA).

### 2.7. Quantitative Analysis of Adipokines, Chemoattractants, and Growth Factors

Secretion of adipokines, chemoattractants, and growth factors were analyzed from cell culture supernatant after 5 days of cell culture in the presence and absence of oxLDL. Custom bead-based multiplex immunoassays were used to quantify secretome (Biolegend, Inc.), following the manufacturer’s instructions. Samples were analyzed using Attune NxT Flow Cytometer equipped with autosampler (Thermo Fisher Scientific, Inc.). Data analysis was done using LEGENDplex™ Data Analysis Software (Biolegend, Inc.).

### 2.8. Gene Expression Analysis

Gene expression analysis was performed using an established protocol, as published [25]. Briefly, total RNA was extracted using TRIzol (Invitrogen, Thermo Fisher Scientific, Inc.) and converted into complementary DNA (cDNA) using a high-capacity cDNA reverse transcription kit (Applied Biosystems, Thermo Fisher Scientific, Inc.). The cDNA concentration and purity (the ratio of absorbance at 260 nm and 280 nm) were quantified using nanodrops (Thermo Fisher Scientific, Inc.) prior to performing gene expression analysis. The primers used in this study were synthesized from Bioneer Inc. The primer sequences are listed in Appendix A. qPCR was performed using the SYBR Green PCR Master Mix (Applied Biosystems, Thermo Fisher Scientific, Inc.). The qPCR procedure was set as follows: denaturation for 5 min at 95 °C; 45 cycles of denaturation (95 °C, 15 s), annealing under primer-specific conditions (30 s), and target gene-specific extension (30 s at 72 °C). Fluorescence signals were measured for 20 s at 72 °C. To confirm the specificity of the PCR products, a melting curve analysis was performed at the end of each run. The Beta-actin gene was used as a reference gene. Experiments were performed in four replicates.

### 2.9. Data and Statistical Analysis

Unless otherwise stated, all experiments were performed in at least four repeats, and data are represented as mean ± standard deviation (SD). Statistical significance has been determined by two-way ANOVA followed by Tukey’s test using Prism 8 (GraphPad Software), and the level of significance was set to *p* < 0.05, unless otherwise stated.

## 3. Results

With atherosclerosis being the main cause of cardiovascular diseases [1], there is an increasing need for platforms that can mimic in vivo conditions. Furthermore, these platforms must be thoroughly characterized, and a profound understanding of individual contributions of the cells involved in atherosclerosis development must be reached before venturing to co-culture models. Most atherosclerosis studies are carried on 2D tissue culture plates [16]. While meaningful contributions have been made towards 3D modeling atherosclerosis in vitro, most studies have focused on co-culture of endothelial cells (ECs), smooth muscle cells (SMCs), and monocytes, in either physiological [18] or synthetic [20] scaffolds, while detailed individual contributions of macrophages still remain unclear.

Here, we embedded monocytic THP-1 cells into 3D type I collagen matrices of low and high concentrations, surrogates for the plaque of low and high densities. As mentioned earlier, collagen I is known to make up for approximately two-thirds of the total collagen in atherosclerotic plaques [22], and the progression of atherosclerosis leads to an increase in collagen concentration within the different layers of diseased vessels. By reconstituting collagen matrices at low and high densities, we were able to mimic the changes of the ECM at early and advanced stages of atherosclerosis. The pore size of the collagen matrices was 11.54 ± 0.88 µm and 3.67 ± 0.95 µm for 1 mg/mL and 3 mg/mL collagen concentration, respectively, and as expected, matrix elasticity was enhanced with an increase of collagen concentration and was 52.10 ± 10.38 Pa and 211.43 ± 15.62 Pa for 1 mg/mL and 3 mg/mL collagen concentration, respectively. The resulting pore sizes span the pore sizes found in vivo [28,30]. Those of atherosclerotic plaque have not been experimentally measured as far as we know, likely due to the complexity of such tissues. However, it has been reported that regressive plaques (i.e., more advanced) have an increased content of collagen [33]. Embedded cells were differentiated into macrophages and further activated into distinct pro- and anti-inflammatory phenotypes. Cells were cultured in the presence of oxLDL simulating atherosclerosis conditions in vitro for 5 days. The duration of the treatment was experimentally determined by measuring oxLDL uptake at different timepoints, we observed that engulfment of LDL by all cells plateaued from 5 days onwards (data not shown). To understand the differences in response by each cell type, we evaluate the uptake of lipids, further assess their expression of relevant oxLDL receptors, quantified their secretomes, and finally examined their expression of selected surface markers to discern if there are any phenotypic changes due to oxLDL exposure.

### 3.1. %oxLDL + Cells Are Greatest in Monocytes and M1 Accumulates the Most oxLDL

Monocytes and macrophages were cultured in the presence and absence of DiI fluorescent-tagged oxLDL (DiI-oxLDL) for 5 days. As shown in Figure 1A, it can be observed that macrophages uptake a higher amount of oxLDL than monocytes, and particularly M1 appears to engulf more oxLDL than other macrophage phenotypes. No differences in uptake could be visually observed between low and high matrix density for the same cell phenotypes. To confirm the visual impressions, cells were harvested from 3D collagen matrices by digestion with collagenase and oxLDL uptake was analyzed by flow cytometry. The percentage of oxLDL positive cells (oxLDL+ cells; cells that uptake oxLDL) and the uptake capacity were assessed using mean fluorescence intensity (MFI). As shown in Figure 1B, we observed a significantly high percentage of monocytes that uptake oxLDL when compared to macrophages (see Appendix A). However, monocytes engulf a lower amount of oxLDL, as demonstrated through MFI. M1 showed significantly higher uptake compared to M0 and M2 macrophages (Appendix A) but without significant difference across matrix densities. We further quantified the size of lipid droplets using a custom-built image analysis toolbox written in MATLAB. Interestingly, we found larger lipid droplets in low matrix density in all macrophage phenotypes, while no differences could be observed in monocytes.

In sum, our results suggest higher oxLDL uptake in all macrophages, especially in M1 phenotypes, than in monocytes. This finding is in line with the high phagocytic activity of M1 macrophages [34], while it contradicts reports that showed M2 macrophages have a higher capacity to accumulate LDL than M1 in the 2D culture model [35,36]. Literature reports that M1 macrophages are the predominant phenotype present in human and mouse plaque lesions and directly contribute to atherosclerotic plaque formation in vivo [11,37,38,39,40], which supports our findings in the 3D cell culture model. Both %oxLDL+ cells, as well as lipid size, was reduced in higher density matrices.

### 3.2. oxLDL Induced LOX1 Expression in Monocytes and Macrophages, while It Enhanced CD36 Expression in M2 Phenotypes

To better understand the oxLDL uptake and lipid size formation, we assessed if the presence of oxLDL affects its uptake receptors. We analyzed the gene expression profile of two main receptors responsible for oxLDL uptake, namely lectin-like oxidized low-density lipoprotein receptor-1 (*LOX1*) and scavenger receptor class B member 3 (*CD36*) through qPCR [41,42]. LOX1 is a transmembrane glycoprotein that binds to and internalizes oxLDL resulting in distinct cell-type-specific downstream signaling, which leads to differential cellular behaviors [43]. As shown in Figure 2A, in the absence of oxLDL, no significant change in LOX1 could be observed between monocytes and macrophages as well as between matrix density. After treatment with oxLDL, all cells exhibited up-regulation of LOX1 expression with the highest expression in M1, tying in with our MFI data, followed by M2, M0 macrophages, and then monocytes. Our finding is corroborated by other reports suggesting an enhanced LOX1 expression via activated nuclear factor kappa-light-chain-enhancer of activated B cells (NF-kB) [44], pro-inflammatory cytokines mediated LOX1 expression [45], as well as oxidative stress [46]. The mentioned mechanisms might also explain the higher expression of LOX1 in M1 phenotypes. Contradicting our MFI data, we found slightly higher expression in the dense matrix in monocytes and macrophages, except M2 phenotypes. We postulate that this observation might be caused by an enhanced modulation of pro-inflammatory response by macrophages in a dense matrix, as previously reported [25]. Further evidence that collagen concentration can modulate the phenotype of macrophages is found in atherosclerotic plaque progression and regression studies [47,48,49], where progressive plaques (lower collagen content) contain more pro-inflammatory macrophages, while more anti-inflammatory macrophages are found in regressive plaques (higher collagen content).

CD36 is a plasma membrane glycoprotein that binds a diverse array of ligands, including oxLDL [50,51]. As shown in Figure 2B, we found higher CD36 expression solely in M2 macrophage in the absence and presence of oxLDL, whereby a significant up-regulation of CD36 expression could be observed after treatment with oxLDL. The expression of CD36 in THP-1 derived M2 has been reported; these cells were polarized based on a similar protocol [52]. Although M2 showed an increase in both LOX1 and CD36 expression, the amount of oxLDL uptake is lower than the M1 counterpart, which exhibited up-regulation of LOX1. The reason could be on account of M2 being anti-atherosclerotic [13] or because of LDL degradation after CD36-mediated uptake [51]. Although the contribution of LOX1 in the oxLDL uptake is reported to be minimal compared to other receptors [42], an increase of soluble LOX1 can contribute up to 40% in oxLDL uptake [43]. Similar to LOX1, we observed a slight increase in CD36 expression of M2 in the denser matrix after treatment with oxLDL. A report suggested a correlation between vessel stiffening and CD36 expression in endothelial cells [53]. To further understand the contributing role of monocytes and macrophages within the atherosclerotic lesion, we performed a quantitative analysis of their secretome using custom bead-based multiplex ELISA.

### 3.3. RBP4 Is Up-Regulated by All Cell Types with the Addition of oxLDL and Enhanced by High-Density Matrices for THP-1_+LDL_ and M1_+LDL_

Adipokines retinol-binding protein 4 (RBP4), resistin, and leptin are reported to support the progression of atherosclerosis, while adiponectin has protective actions against plaque formation [54]. These factors are also known to be released along with other pro-inflammatory factors during the development of obesity [55,56,57], and within adipose tissue, they have been shown to induce inflammation by activating macrophages [58]. While no significant statistical differences were found in adipokines, some distinct qualitative trends were observed, which are discussed below. Clinically, RBP4 is a marker for atherosclerotic-associated cardiovascular disease [59]. In animal models, RBP4 has been found to positively contribute towards atherosclerotic progression and mediate LDL uptake [60]. In our model, we found an increase in RPB4 secretion by all cell types in response to oxLDL (Figure 3A); generally, higher secretions were measured in the denser matrix for THP-1_+LDL_ and M1_+LDL_. It has been demonstrated that RBP4 expression was enhanced in the areas rich in macrophage foam cells in atherosclerotic lesions of aortic specimens from both humans and apolipoprotein E-deficient mice [60], suggesting foam cell-like formation in our biomimetic atherosclerosis models. 

Leptin reportedly exacerbates inflammation [61] in low matrix density; our results showed that THP-1_+LDL_ and M0_+LDL_ reduced leptin secretion (Figure 3B) as with resistin (Figure 3C). It is reported that human macrophages secrete resistin on their own, and this contributes to atherosclerosis development directly by causing endothelial and smooth muscle cell (SMC) dysfunction [62,63], potentially increasing the permeability of LDL into adjacent tissue. Only THP-1_+LDL_ showed elevated leptin secretion in denser matrices. Evidence suggests that leptin and resistin may initiate the recruitment of monocytes, macrophages, endothelial cells, and smooth muscle cells towards the atherosclerotic site [64,65]. At site, these cell types all individually progress atherosclerosis, although beyond the scope of this study, co-culture experiments with the above-mentioned cell types with our biomimetic model have to be performed to enhance the understanding of this complex disease. The atheroprotective adiponectin was below the assay’s theoretical limit of detection using the bead-based ELISA kit (minimum detectable concentration 41.1 pg/mL) in all cells with and without treatment with oxLDL, potentially due to its short half-life [66]. Overall, with the addition of oxLDL there is an increase of secreted adipokines in the dense matrix, particularly by THP-1_+LDL_ and M1_+LDL_, similarly to what would be expected in an advanced atherosclerotic plaque.

### 3.4. In High-Density Matrices THP-1_+LDL_ and M1_+LDL_ Secretion of Immune Cell Chemoattractants Are Elevated and Suppressed in M2_+LDL_

We measured IP-10, MCP-1, and IL-8 levels from our cell culture system as these are known chemoattractants that are involved in the recruitment of inflammatory cells to the atherosclerotic site [67,68,69]. Both IP-10 (Figure 4A) and IL-8 (Figure 4B) concentrations are significantly elevated in the dense matrix with M1_+LDL_ compared to the other cell types, whereas the chemoattractants are suppressed with M2_+LDL_. IL-8 recruits cells that participate in acute inflammation [70], while IP-10 is a chemoattractant for immune cells from the adaptive immune system to the atherosclerotic site [71]. It is reviewed in the literature that the adaptive immune system during atherosclerosis changes from a protective to a pathological response, typically when the ratio of effector T-cells to regulatory T-cells increases [72,73]. Therefore, the increased secretion of both IL-8 and IP-10 by M1_+LDL_ in denser collagen matrices, and hence the respective immune cell types, is in response to the stiffer tissue in advanced atherosclerotic plaques.

Unanticipated is the elevated secretion of MCP-1 (Figure 4C) by M2. It is qualitatively highest compared to the other cell types, elevated in low-density matrices and suppressed in denser matrices. Interestingly, both IP-10 and IL-8 in our experiment are secreted at similar concentrations as those measured from atherosclerotic plaques ex vivo [74]. Particularly, IP-10 plays an anti-atherogenic role by inhibiting angiogenesis [75,76]. M2_+LDL_ up-regulation of IP-10, especially in low-density collagen, could signify its potential anti-atherogenic role.

### 3.5. PDGF-AA, EPO, and M-CSF Are Up-Regulated by Macrophages and VEGF Is Up-Regulated by Monocytes

Endothelial progenitor cells (EPCs) play an important role in the recovery and repair of adult vasculature [77,78]. EPCs reside in bone marrow (BM) and can be signaled to go into circulation to contribute towards the generation of new blood vessels or repair of damaged ones. As atherosclerosis is a result of initial and continual structural and functional endothelial damage, it is expected that EPCs will have an influence on the disease. Clinically, it was observed that high EPC count is a negative predictor of the occurrence of atherosclerotic plaque [79]. Furthermore, recruitment and incorporation of endothelial progenitor cells endogenously into atherosclerotic sites have been shown to attenuate the progression of the disease and mediate vascular repair [80]. For these reasons, we wanted to determine if macrophages within the atherosclerotic microenvironment secrete growth factors to facilitate vascular repair (Figure 5). While differences in growth factor secretion can be observed between cell types, and between treated and untreated cells, no matrix dependence has been observed. There were no significant differences in any of the growth factors, however, the qualitative trends discussed in this section can be discerned from the data.

Monocytes and all macrophages produce elevated PDGF-AA levels when exposed to oxLDL (Figure 5A). PDGF is known to be involved in the development of new blood vessels [81], and in a recent study, using a mice model, it has been shown that PDGF-AA has the ability to enhance the healing effects of EPCs [82].

The role of VEGF in atherosclerotic plaques is contradictory. Measurement of increased microvessel density in ruptured and unruptured human plaques suggests that VEGF, and other factors that promote blood vessel development, advances atherosclerosis towards destabilization [83]. VEGF is also known to induce the growth of pre-existing vessels as well as to promote the growth of new blood vessels, and therefore this growth factor is also key in vessel repair [84]. From our experiments, oxLDL+ polarized macrophages and monocytes exhibited up-regulation of VEGF secretion (Figure 5B).

EPO stimulates the mobilization of EPCs from bone marrow, animal models, and patients treated with EPO routinely result in increased EPCs in circulation [85,86]. EPO is postulated to also promote homing and differentiation of EPCs at sites of vascular injury [86]. From our results, EPO levels are consistently elevated with the application of oxLDL on macrophages, regardless of their subtype or the matrix density (Figure 5C). EPO, which may inhibit foam cell formation in mice [87], is secreted equally by all our macrophages, whether it has an inhibiting role in our model will need to be further investigated, but initially it could again imply an auto-regulating function of these cells.

M-CSF secretion, like PDGF-AA and EPO, is reserved for macrophages (Figure 5D). Although it has no direct involvement with EPCs, M-CSF has multiple roles in the atherosclerotic microenvironment by regulating migration and recruitment of monocytes and also survival and scavenger activity of macrophages [88]. Although M-CSF is known to induce macrophages into the anti-inflammatory phenotype M2 [89], we observed secretion of this growth factor by M0_+LDL_ and M1_+LDL_, which again could suggest a role in auto-regulating atherosclerosis.

### 3.6. oxLDL Enhances Expression of CD68 in Monocytes and MHCII Is Enhanced in Dense Matrices for All Cell Types

The consensus is that recruited monocytes will get activated in response to oxLDL into pro-inflammatory macrophages during atherosclerosis [1], we, therefore, analyzed gene expression of *CD68* and *MHCII*. CD68 is a pan-macrophage marker and is up-regulated upon macrophage differentiation [90,91,92], while MHCII was found to be a marker for pro-inflammatory macrophages [25]. As expected, without oxLDL, as compared to monocytes, we found higher CD68 expression in all macrophages (Figure 6A). In the presence of oxLDL, we found an up-regulation of CD68 in monocytes, which reaffirms the theory of monocyte differentiation into macrophages by oxLDL. No change in CD68 expression could be observed in macrophages with or without oxLDL. This contradicts reports demonstrating the up-regulation of CD68 upon oxLDL treatment in mouse bone-marrow-derived macrophage [93] and also in THP-1-derived M0 macrophages [94]. The discrepancy in CD68 up-regulation upon oxLDL treatment may arise from cell culture dimensionality. Our 3D matrix density did not have any impact on CD68 expression.

As we found higher pro-inflammatory adipokine secretion, we hypothesize that oxLDL is capable of influencing a pro-inflammatory macrophage phenotype, which we ascertain through the analysis of the *MHCII* gene expression. As shown in Figure 6B, MHCII is highly expressed in M1 macrophages with higher expression in the dense matrix as corroborated by our previous work [25]. In the presence of oxLDL, monocytes and all macrophages presented enhanced *MHCII* gene expression with the highest in M1. This is supported through the reported oxLDL-mediated uptake activation of spleen tyrosine kinase (SYK) that, in turn, up-regulates MHCII expression in macrophages [95]. Our results suggested that the enhancement of vascular stiffening via matrix density might increase the pro-inflammatory phenotype in response to oxLDL and thus contribute to the progression of atherosclerosis.

## 4. General Discussion and Conclusions

As mentioned previously, there is a necessity to understand the neglected biochemical contributions by monocytes and macrophages to the atherosclerotic microenvironment. Comprehending the adipokine and growth factor contributions by the various immune cell types within the microenvironment could lead to de novo therapeutic strategies against atherosclerosis. We first revealed that adipokines and growth factors are indeed found in our in vitro model for atherosclerosis. THP-1 monocytes and their derived macrophage subtypes exhibited distinct oxLDL uptake, gene expression, and secretome. As oxLDL and lipid size have no significant differences between cells embedded in low or high matrix densities, we can conclude that several of these differential biological readouts are mediated by the density of the collagen hydrogel-based matrices.

From an adipokine point of view, THP-1_+LDL_ and M0_+LDL_ seem to have a better role in suppressing atherosclerosis in our model, whereby there is down-regulation in adipokines. This is observed predominantly in less dense collagen, which would correspond to earlier stages of atherosclerosis. Moreover, the higher amounts of adipokines in denser matrices reinforces that our denser matrix model produces a cellular reaction similar to more advanced atherogenic stages. In all, with our cell-line-based model of atherosclerosis, adipokines are measurable, and these secretions should be made aware of as they have shown to have an influence on the pro-inflammatory potency of the immune cells in an autocrine manner. In terms of immune cell recruitment, adaptive immune cells are recruited by all macrophage types, with M1_+LDL_ having the greatest influence on their influx into atherosclerotic sites. M2_+LDL_ and M1_+LDL_ actively recruit innate immune cells to atherosclerotic sites, via MCP-1 and IP-10, respectively, also in a matrix density-dependent manner. Finally, the up-regulation of the pro-inflammatory marker MHCII and pan-macrophage marker CD68 by THP-1, confirm that monocytes are differentiating into macrophages in the presence of oxLDL. Furthermore, MHCII shows a trend with collagen concentration, where it is higher in denser collagen matrices. The mechanisms of this matrix-regulation on the cells require further experimentation and are not currently addressed through our study.

The role of M2 is controversial in atherosclerosis, with some studies claiming it could support plaque regression [13], while others argue it supports progression by polarizing towards a pro-inflammatory phenotype instead [35]. Our results suggest that after the removal of activating media and the addition of oxLDL, M2 may adopt a relatively more pro-inflammatory phenotype, in spite of its lower level of oxLDL uptake. This is especially the case in lower density matrices, while in higher density, it seems to retain more of its original phenotype. It is worth noting that after the removal of polarizing media, macrophages are no longer forced into their intended M1 or M2 phenotype. Instead, they are malleable [96], straddling between the M1 and M2 boundaries according to their specific responses to oxLDL and their microenvironment. The same applies to M1 as well as M0, because of their plasticity, their phenotypes are dynamic and non-binary [25], making investigations into atherosclerosis complex, as observed in this study.

From a tissue density perspective, our model represents two different stages of atherosclerotic plaque progression, where a high-density matrix would correspond to a more advanced stage than low density. The main effects of these two different stages of atherosclerosis on each cell type in this study are depicted in Figure 7. Notably, M1_+LDL_ displayed a more pro-inflammatory phenotype in high tissue density, as it is deduced by their secretome and pro-inflammatory marker expression. While THP-1_+LDL_ suppressed adipokine secretion in low tissue density, the same trend was not found in high tissue density, where RBP4 was greatly increased, which, together with an up-regulation of MHCII, suggests that as atherosclerosis progresses, these cells may adopt a more pro-inflammatory behavior. Moreover, although growth factors were secreted, no matrix dependence was observed. As mentioned above, the role of M2_+LDL_ is somewhat controversial, however, when compared to the other cell types in our model, it appears to adopt a slightly more pro-inflammatory role at the early stages of atherosclerosis supporting plaque progression, and becomes more anti-inflammatory as the atherosclerotic plaque advances, as it is surmised by a decrease of RBP4, IP-10 and MCP-1 in high-density matrices.

In conclusion, we have analyzed the oxLDL uptake, secretome, and phenotype of monocytes and macrophages when placed in a physiologically relevant ECM. We have shown that adipokines and growth factors are, in fact, secreted by macrophages and must be taken into considerations for research into atherosclerosis regression and repair. Moreover, our cell-line-based model is able to differentiate monocytes into macrophages after treatment with oxLDL, and these macrophages showed reduced phagocytic capacity and up-regulation of VEGF, which are traits also found in Mox, anti-oxidant macrophages found in murine atherosclerotic plaques, which is induced by oxLDL [97,98]. Future studies to expand this model should include other relevant cells in atherosclerosis, such as SMC and EC, moreover, whether THP-1 derived macrophages have the same phenotype as Mox will also need to be confirmed.

## Figures and Tables

**Figure 1 bioengineering-07-00113-f001:**
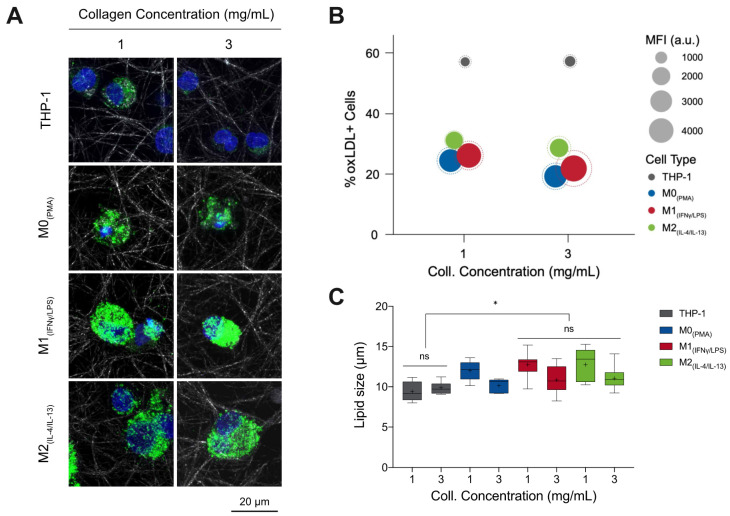
oxLDL uptake by monocytes and macrophages in 3D collagen matrices. (**A**) Representative confocal images of monocytes and macrophages with DiI-oxLDL uptake. (**B**) In this infographic, the number of oxLDL+ cells (*y*-axis) was greatest in THP-1 (grey) followed by M2 (green) macrophages, then M1 (red), and finally M0 (blue). The amount of oxLDL in the cells, as quantified through mean fluorescence intensity (MFI, solid circle) (circle size, see legend), was greatest in M1 macrophages, followed by M0, then M2, and finally THP-1. The dotted circle is the corresponding standard deviation of the MFI. A sample size of 5 was performed for each condition. (**C**) Lipid droplet size as measured from images obtained with confocal microscopy using a custom-built image analysis toolbox. Data were shown as a boxplot (whiskers represent minimum and maximum, + represents the mean, line inside each box plot represents the median). Significance is represented as * *p* < 0.05 between samples in 1 mg/mL collagen concentration.

**Figure 2 bioengineering-07-00113-f002:**
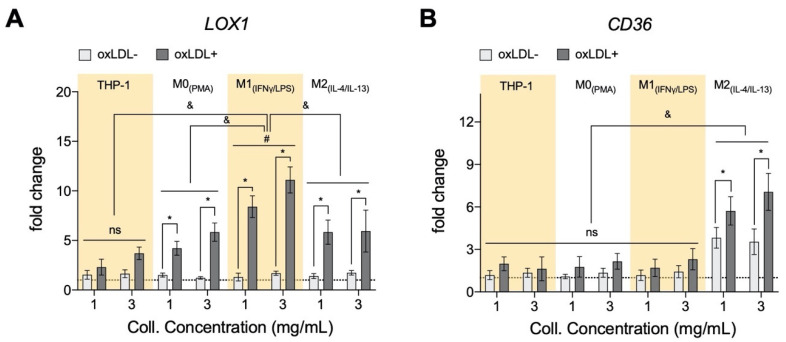
Expression of oxLDL receptors of monocytes and macrophages cultured in 3D collagen matrices. (**A**) There is an increase in gene expression of lectin-like oxidized low-density lipoprotein receptor-1 (*LOX1*) in both monocytes (THP-1) and significantly in all macrophage subtypes when cells are cultured in 3D collagen hydrogels and exposed to oxLDL. Significantly, M1_+LDL_ exhibited the highest fold induction and THP-1_+LDL,_ the lowest, with M0_+LDL_ and M2_+LDL_ comparable. LOX1 expression only changed significantly in M1 with collagen concentration (*x*-axis, 1 mg/mL to 3 mg/mL). (**B**) Genetic expression of *CD36*, another oxLDL receptor, was only measurable in M2_+LDL,_ and significantly higher expression was found in lower density collagen hydrogels. All significances represented as *p* < 0.05, * compared to untreated sample at the same matrix condition, & between oxLDL samples of different cell types at the same matrix condition, and # significant change with collagen concentration. The dotted horizontal line indicates a level of no change.

**Figure 3 bioengineering-07-00113-f003:**
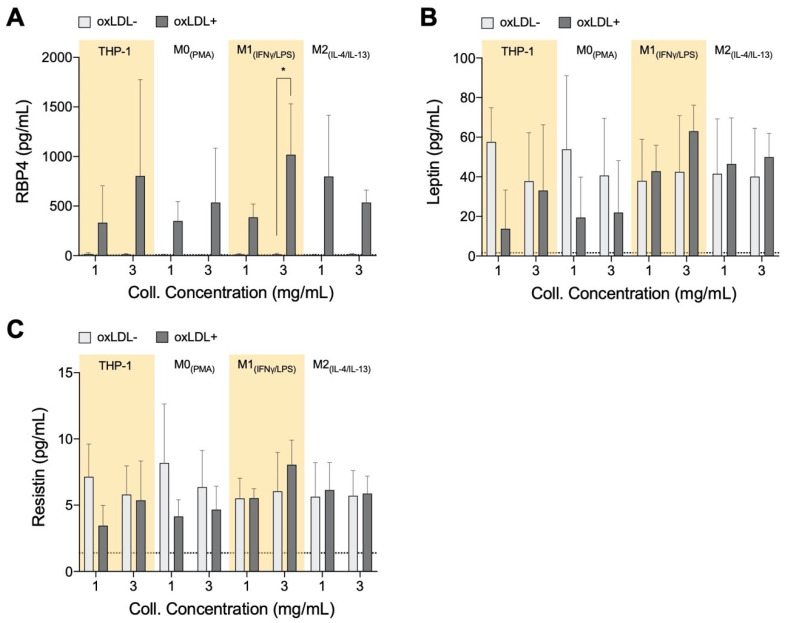
Adipokine secretome from monocytes and macrophages cultured in 3D collagen hydrogels. Box plots of absolute secretion levels of (**A**) RBP4, (**B**) leptin, and (**C**) resistin to observe trends and indicate comparisons that are statistically significant. The dotted line in the box plots marks the lowest detectable concentration of assay. Statistical significance has been determined by two-way ANOVA followed by Tukey’s test, and the level of significance is represented as * *p* < 0.05, significantly higher than the untreated sample at the same matrix density condition, no significance otherwise.

**Figure 4 bioengineering-07-00113-f004:**
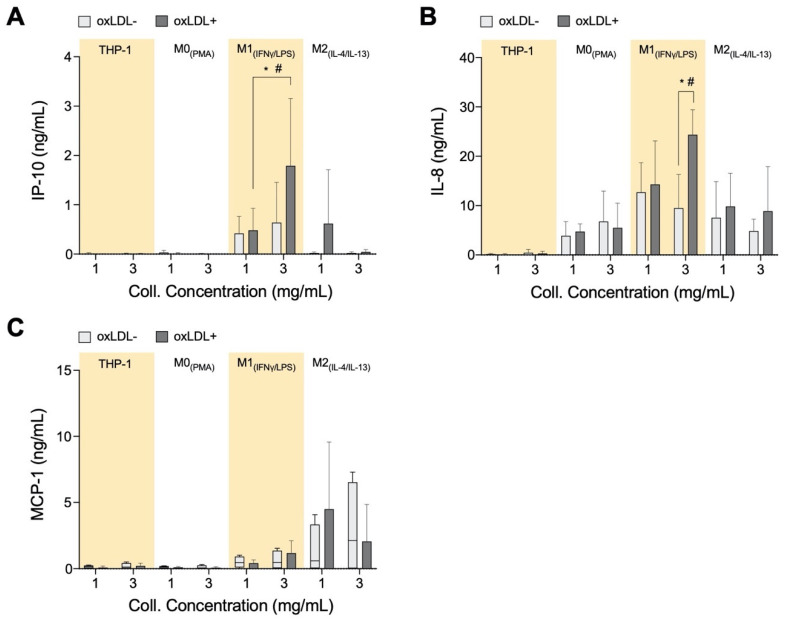
Chemokine secretome from monocytes and macrophages cultured in 3D collagen hydrogels. Box plots of absolute secretion levels of (**A**) IP-10, (**B**) IL-8, and (**C**) MCP-1 to observe trends and indicate comparisons that are statistically significant. The dotted line in box plots marks the lowest detectable concentration of assay. Statistical significance has been determined by two-way ANOVA followed by Tukey’s test, and the level of significance is represented as * *p* < 0.05 and # *p* < 0.05 in 3 mg/mL between THP-1_+LDL_, M0_+LDL,_ and M2_+LDL_
*vs.* M1_+LDL_. The dotted horizontal line indicates the lowest detectable limits of the assay.

**Figure 5 bioengineering-07-00113-f005:**
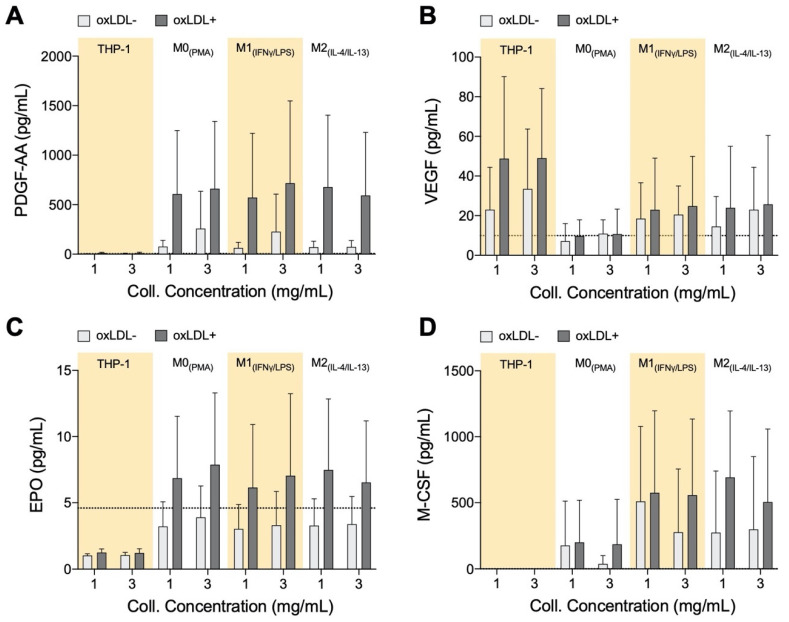
Growth Factor Secretome of monocytes and macrophages cultured in 3D collagen hydrogels. Box plots of absolute secretion levels of (**A**) PDGF-AA, (**B**) VEGF, (**C**) EPO, and (**D**) M-CSF to observe trends. The dotted line in the box plots marks the lowest detectable concentration of assay. Statistical significance has been determined by two-way ANOVA followed by Tukey’s test, and the level of significance was set to *p* < 0.05.

**Figure 6 bioengineering-07-00113-f006:**
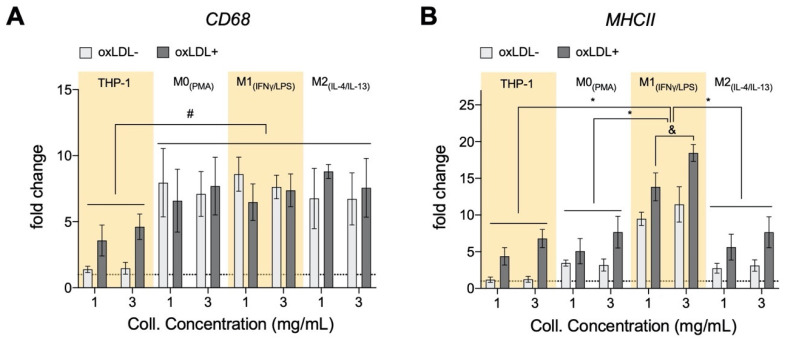
Expression of selected markers of monocytes and macrophages cultured in 3D collagen hydrogels. (**A**) For all cell types, the pan-macrophage marker (CD68) was also increased with the exposure to oxLDL, with THP-1 having the highest fold induction. (**B**) There was an increase in gene expression of the well-acknowledged activation maker (MHCII) in all cell types after exposure to oxLDL. There is also a general increase in expression with increment of collagen hydrogel density (*x*-axis, 1 (1 mg/mL) → 3 (3 mg/mL)). This correlation of MHCII gene expression with tissue density is statistically significant for THP-1 and M1. All significances represented as *p* < 0.05, * compared to corresponding untreated and treated samples at the same matrix condition, and significant change with collagen concentration and # significant compared to untreated samples at the same matrix conditions. The dotted horizontal line indicates a level of no change.

**Figure 7 bioengineering-07-00113-f007:**
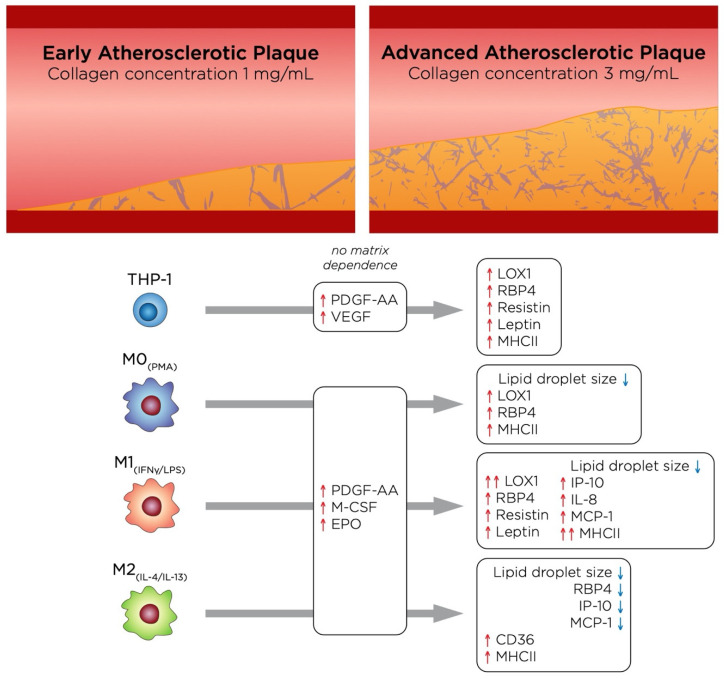
Illustration summarizing the main findings of this study, which highlights the changes undergone by the cells as atherosclerosis progresses.

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
