# Peer review of "Biomimetic 3D Models for Investigating the Role of Monocytes and Macrophages in Atherosclerosis"

_bioengineering, 2020, doi:10.3390/bioengineering7030113_

Round 1

Reviewer 1 Report

  1. The experiment investigated two concentration of collagen, namely 1 mg/ml and 3 mg/ml. Is this an arbitrary choice or values that are back up through clinical interrogation? To see the impact of ECM stiffness on the progression of atherosclerosis It would be more illuminating if the investigators would explore more variations in terms of the concentrations. Any reason why only two concentrations and not more were explored?
    1. We are seeing in this current paper the extreme, or the start (low densities) and late (high densities) stage of the plaque formation. However, what connect the two ends, i.e the progression of the disease per se, are still considered important and is not thoroughly covered/investigated in the current study.
  2. The accumulation of foam cells may play a significant role in the rigidity of the surround ECM. Have the authors also assessed the contributions of these foam cells, as opposed to only the collagen ECM?
  3. Any reason why we culture and observe at end of 5 days? Why not 7 days or 30 days?
  4. The current study explored the contribution of monocytes and macrophages to the diseases progression, but not the contribution of dendritic cells/SMC to the progression of disease.

Reviewer 2 Report

The premise of this manuscript has been to test a new 3D culture model for monocytes and macrophages that mimics the atherosclerotic plaque environment. The authors use 2 different collagen concentrations in which to grow THP-1 cells. THP-1s are a human acute monocytic leukemia cell line, often regarded as pre-monocytic in their dividing undifferentiated state. The authors have used PMA to differentiate the cells to become monocytic/macrophage like, and use pro-inflammatory (LPS/IFNg) or anti-inflammatory (IL-4/IL-13) stimuli to differentially polarise these further. The activity of these cells with respect to lipid uptake and adipokine, chemoattractant and growth factor secretion was then assessed, whilst grown in 3D collagen cultures. The principles behind the work described in this manuscript are of interest, as there is a need to develop 3D cultures to improve their biomimetic relevance.

The methodology presented here is interesting, and in theory suited to this journal. However, there are some fundamental limitations and over-interpretations to the work presented:

  1. I would like to see more detail in relation to the collagen matrix. The data for pore size and elasticity should be presented. Some microscopy images to visualise differences would also be more convincing particularly as Fig 1A do not look any different with respect to collagen fibres. More details on these measurements should be included in the methods. The explanation as to why this study has been performed has not been well set out in the introduction or abstract.
  2. Results on secretomes are over-interpreted as mostly these are based on non-significant differences. Figs 3, 4 and 5 parts A and B show “heat-maps” for relative changes to secretion products, shown as up- or down-regulated, relative to the base-line controls stated. The actual data these heat-maps are based upon are shown in part C of each of these figures. However, it is clear that this data has a large level of variability, and most differences that are depicted in the heat-maps in are not statistically significantly different. Therefore, these non-significant differences should not be depicted in the heat-maps as this is over-interpretation. In the same way, many of the results are described based on the non-significant changes indicated in the heat-maps. This is over-interpretation and such statements in both the results and discussion are not valid.
  3. Comparisons with no collagen or 2D cultures for these same THP-1 cells and polarisation conditions were not made. Many statements relating to the 3D culture results make comparisons to the literature, rather than using these very relevant controls in order to make such inferences relating to the responses measured. Other literature quoted has not necessarily used THP-1 cells, nor the same PMA treatment (concentration and timings) nor the same polarisation conditions. Therefore, these comparisons are somewhat tenuous and speculative. I would have liked to have seen the no collagen controls for all of these experiments in order to understand the impact a 3D collagen culture has had on the measured responses.
  4. It would have been interesting to have compared this model with primary human blood monocyte-derived macrophages. THP-1 cells are limited in their direct relevance, although I accept this paper is very much about generating a new model where this can be expanded, but I would have liked to see this limitation acknowledged in the discussion at a minimum.

Minor comments on the manuscript:

  1. Statistics: there is a lack of statistic symbols in some of the graphs making the interpretation of results more complicated. The statistical test used is 2-way ANNOVA, which takes into account the effect of both variables (treatment and matrix density). However, the text mostly describes effects of only oxLDL or only matrix density on the cell responses; perhaps for these comparisons where only one variable is taken into account a different test would be more appropriate.
  2. Lines 254-256 “ We postulate that this observation (increased LOX1 in M1 macrophages and not in M0 or M2 under high collagen conditions) might be caused by an enhanced modulation of pro-inflammatory response by macrophages in a dense matrix, as previously reported [25]”à self-citation. However, other literature suggests that less stable and progressive plaques with lower collagen content contain increased pro-inflammatory macrophages while more stable or regressive plaques with higher collagen content contain more anti-inflammatory macrophages

https://www.frontiersin.org/articles/10.3389/fimmu.2014.00579/full

https://www.ahajournals.org/doi/10.1161/ATVBAHA.120.314252

https://www.ncbi.nlm.nih.gov/pmc/articles/PMC5906181/

This should be discussed.

  1. Lines 308-310 states “adiponectin was below the assay’s theoretical limit of detection using the bead-based ELISA kit” – I therefore suggest this data should be removed as it is not adding any information to the manuscript.
  1. The results show no changes in growth factor secretion due to collagen matrix density; this is not discussed, despite being a relevant finding
  1. This study focused on oxLDL uptake, which occurs in the early stage of plaque formation after the collagen and the cap are produced. It would interesting to use this collagen matrix to study the role of macrophages in plaque stability (secretion of MMPs, degradation of collagen, proteoglycans) or to study macrophage apoptosis, mimicking what happens in necrotic cores. Phagocytosis and foam cell formation also occurs in advanced plaques but usually in distal areas from the necrotic cores where the matrix collagen is found. As for major point 1 above, it would be interesting to view more images of the cells cultured in these matrices – does the polarisation state alter the matrix density of the, since pro-inflammatory macrophages can degrade the surrounding collagen? It appears from figure 1 that the M1 in dense collagen have fewer fibres surrounding the cell; at present these are representative figures, but the authors may consider quantifying these images and providing more data to enhance the information in this manuscript.

Round 2

Reviewer 2 Report

The authors have addressed the suggestions raised in my previous review, to a satisfactory level.

The only question where I am not convinced that the change is warranted, is the change of the microscopy images in Figure 1. These have been altered so images are shown at slightly higher magnification. However, this does not really help to show much more in the way of viewing collagen matrices, in fact the original figure 1 is better to view these in this respect. I was hoping for some graphical quantification in the supplementary results where collagen had been measured, although the reviewers have addressed my point in referring me to their previous publications. On balance I suggest re-instating Figure 1 from the original first submission.